# Resolving the spin splitting in the conduction band of monolayer $MoS_2$

Kolyo Marinov[1,2], Ahmet Avsar[1,2], Kenji Watanabe [3], Takashi Taniguchi[3] & Andras Kis [1,2]

Time-reversal symmetry and broken spin degeneracy enable the exploration of spin and valley quantum degrees of freedom in monolayer transition-metal dichalcogenides. While the strength of the large spin splitting in the valance band of these materials is now well-known, probing the 10–100 times smaller splitting in the conduction band poses significant challenges. Since it is easier to achieve n-type conduction in most of them, resolving the energy levels in the conduction band is crucial for the prospect of developing new spintronic and valleytronic devices. Here, we study quantum transport in high mobility monolayer $MoS_2$ devices where we observe well-developed quantized conductance in multiples of $e^2/h$ in zero magnetic field. We extract a sub-band spacing energy of 0.8 meV. The application of a magnetic field gradually increases the interband spacing due to the valley-Zeeman effect. Here, we extract a g-factor of ~2.16 in the conduction band of monolayer $MoS_2$.

[1] Electrical Engineering Institute, École Polytechnique Fédérale de Lausanne (EPFL), CH-1015 Lausanne, Switzerland. [2] Institute of Materials Science and Engineering, École Polytechnique Fédérale de Lausanne (EPFL), CH-1015 Lausanne, Switzerland. [3] National Institute for Materials Science, 1-1 Namiki, Tsukuba 305-0044, Japan. Correspondence and requests for materials should be addressed to A.K. (email: andras.kis@epfl.ch)

Layered semiconducting materials have been extensively studied in the past decade due to their unconventional physical properties[1–4]. One of the recently most studied classes of materials are semiconducting transition-metal dichalcogenides (TMDCs). At the monolayer limit, these materials become direct band gap semiconductors[5–8]. Missing inversion symmetry enables spin splitting at the edges of both valence and conduction bands[9]. More interestingly, as the two degenerate K and K´ valleys are coupled to the two sublattices in the monolayer, they exhibit degenerate band edges with opposite spin orientation leading to the unique presence of spin-valley locking in these materials[2,4,10]. This property is crucial for the investigation of novel spin-valley physics in monolayer TMDCs.

The magnitude of spin splitting has been theoretically well understood by calculating the band structure of these materials via DFT, GW, TB, and other common approaches[3,4,10–12]. At the valence band maximum (VBM), the values range between 150 and 460 meV, while at the conduction band minimum the spin splitting is relatively smaller, predicted to be in the 1–50 meV range. Since the magnitude of spin–orbit splitting is expected to increase with the atomic number, the splitting is smallest in $MoS_2$ which is also the best studied crystal from the TMDC family. Despite the great progress in theory, experimental confirmation of these values is still scarce. One experimental tool allowing the direct access to complete band structures is spin-resolved ARPES or k-PEEM, which allows the imaging of the material band diagram in the reciprocal space under the Fermi level. In this manner, spin splitting in the valence band of TMDCs was experimentally demonstrated[13–15]. However, the smaller spin–orbit splitting in the conduction band could not be resolved due to the energy resolution limit of the method in the range of 20–25 meV[13,15]. A way to circumvent this resolution problem is to study the Fermi surface of $MoS_2$ electrically by investigating the electron transport in the conduction band.

In this work, we experimentally study the strength of spin splitting in the conduction band of monolayer $MoS_2$ by performing quantum transport measurements in the split-gate geometry. We realize a quantum point contact (QPC) and observe conductance quantization with lifted degeneracies, which is then investigated as a function of bias offset and magnetic field. We find that the electron g-factor in the conduction band is $2.16 \pm 0.13$.

## Results

**Device structure**. In our van der Waals heterostructure, monolayer $MoS_2$ is the active channel. It is encapsulated between atomically flat h-BN layers and contacted to multilayer graphene, Fig. 1a. Monolayer $MoS_2$ flakes are first identified by their optical contrast on the substrate. The monolayer thickness is later confirmed by photoluminescence imaging using a dark field optical microscope (see Supplementary Fig. 1 and Supplementary Note 1)[16]. Atomically flat, defect-free h-BN layers are utilized as a high quality substrate and dielectric to preserve the intrinsic electronic properties of $MoS_2$. Few-layer graphene is used as a work-function tunable contact which has been proven to match the work function of monolayer $MoS_2$ for effective charge injection[17]. Source-drain metal contacts and split gates are defined using conventional e-beam lithography followed by e-beam deposition of 2/85 nm Ti/Au electrodes. The distance between the circularly shaped split gates is ~100 nm, allowing the confinement of electrons in the one-dimensional channel. An optical micrograph of the finished device is shown in Fig. 1b and an AFM image of the split-gate geometry is presented in Fig. 1c. Electrical measurements were performed in a pumped variable temperature insert in a helium bath cryostat with a base temperature of 1.4 K. We used the conventional lock-in technique at low frequency of 13

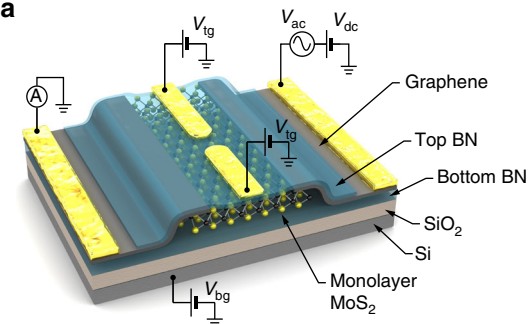

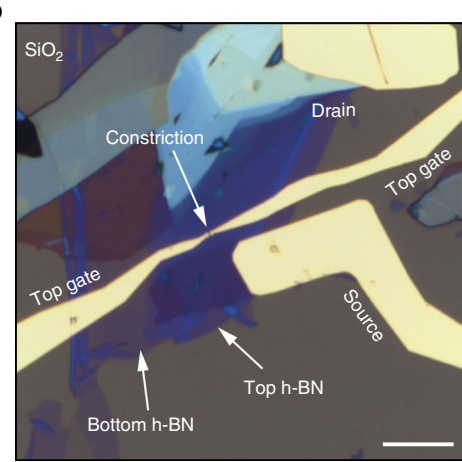

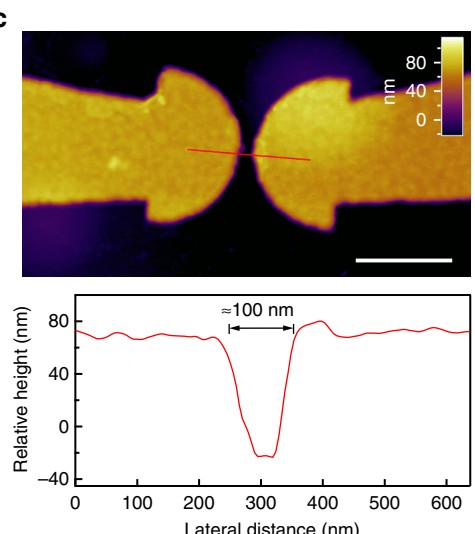

**Fig. 1** Dual-gated encapsulated monolayer $MoS_2$ transistor. **a** The 3D schematics of the device shows all the layers in our device and the metallic electrodes. **b** An optical micrograph of the final device after contact deposition. Scale bar is 10 μm. **c** An AFM image of the split-gate geometry. Scale bar is 500 nm. Following the red line we measure a distance between the two top gates of about 100 nm

Hz and low AC voltage amplitude of $100 \, \mu V < k_B T/e$ to avoid sample heating. Back ($V_{bg}$) and top ($V_{tg}$) gate potentials were applied using a DC bias source. In all presented measurements, the split top gates are biased at the same $V_{tg}$.

**Primary characterization**. Figure 2a shows the $V_{bg}$ dependence of conductance measured at the base temperature of 1.4 K. The

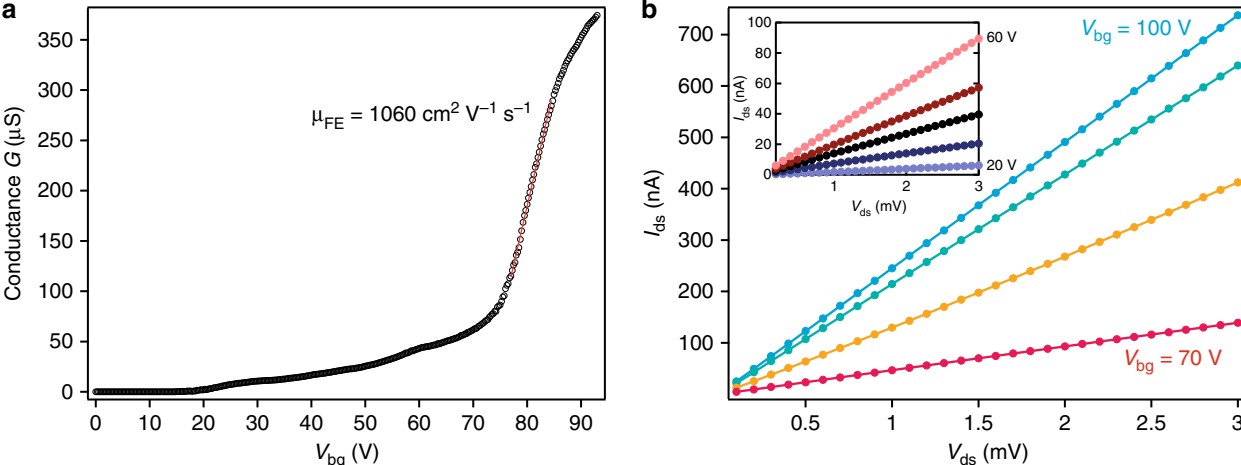

**Fig. 2** Electric performance of the encapsulated monolayer $MoS_2$ with FLG contacts at 1.4 K. **a** Conductance of the device as a function of the back-gate voltage. A sharp turn-on at ~75 V marks the conduction band edge and yields a mobility above 1000 $cm^2\,V^{-1}\,s^{-1}$. **b** Output characteristics of the device at different $V_{bg}$ from 70 to 100 V in steps of 10 V. In all cases, we observe a linear contact characteristic. Inset: current-voltage characteristics in the lower range of $V_{bg}$ from 20 to 60 V in steps of 10 V

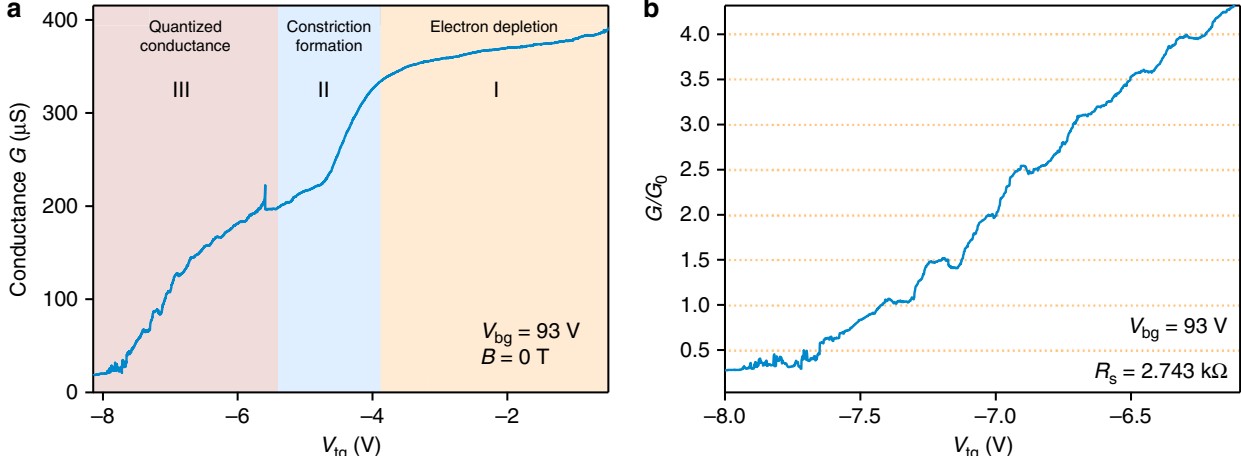

**Fig. 3** Top-gate modulation of the conductance. **a** Raw data of the conductance modulation of the 1L-$MoS_2$ by the split gates at constant high doping induced by the back gate. Three distinguishable regions in the curve correspond to electron depletion under the top gates (I), the formation of a constriction (II) and finally, stepwise turn-off of the device (III). **b** Conductance steps occurring at half and full values of $G_0 = 2e^2/h$ after subtraction of a series resistance $R_s = 2.74\,k\Omega$, indicating the lifting of spin degeneracy due to conduction band splitting

device shows typical n-type behavior. The device conductance turns on near $V_{bg} = 20$ V with monotonically increasing conductance up to 70 V. At around $V_{bg} = 75$ V, we observe a sharp increase in the conductivity followed by a mild saturation of the conductance at $V_{bg} > 90$ V. We assume that the first part of the curve corresponds to the filling of band tail trap states typical for single-layer TMDCs, which was recently measured in capacitance spectroscopy studies[18,19]. The sharp turn-on indicates that the Fermi level is crossing over the conduction band edge. We estimate the field effect mobility of ~1060 $cm^2\,V^{-1}\,s^{-1}$ from the slope of the sharp turn-on. This value is similar to the highest reported values for monolayer $MoS_2$[20]. We have further studied the $I_{ds}/V_{ds}$ characteristics of the device at different applied gate voltages in order to investigate the contacts (Fig. 2b). Even at low doping ($V_{bg} = 20$ V to 50 V) and low temperature, the characteristic is linear, indicating low contact resistance and possibly Schottky barrier-free charge injection[17]. These transparent FLG contacts are necessary for the reliable study of the two-dimensional electron gas (2DEG) in the conduction band of $MoS_2$.

Next, we study the influence of the top gates on the 2DEG at high constant back-gate induced doping ($V_{bg} > 90$ V). We aim for the simultaneous realization of highly doped contact areas (electron reservoirs) with low resistance and the formation of a narrow constricting path for the electron flow between them, which will be the QPC. In Fig. 3a, we present the turn-off curve with the top-gate voltage applied symmetrically to both top gates and at constant high doping induced by the back gate. We can clearly distinguish three regions on this curve (Fig. 3a). (I) While the $V_{tg}$ is tuned from −1 to −4 V, the device conductance is slowly decreasing due to the electron depletion of the 2DEG underneath the top gate electrodes. (II) Near $V_{tg} = −4$ V, we observe a sharp decrease in the conductance indicating the formation of a constriction for the electrons in the gap between the split gates. (III) At $V_{tg} < −5.5$ V, we see the clear quantization of the conductance in steps, which is an evidence for the formation of the QPC. Next, we concentrate on this region and analyze the steps of conductance quantization. In Fig. 3b we present the third region after subtraction of a series resistance of 2.74 $k\Omega$ stemming

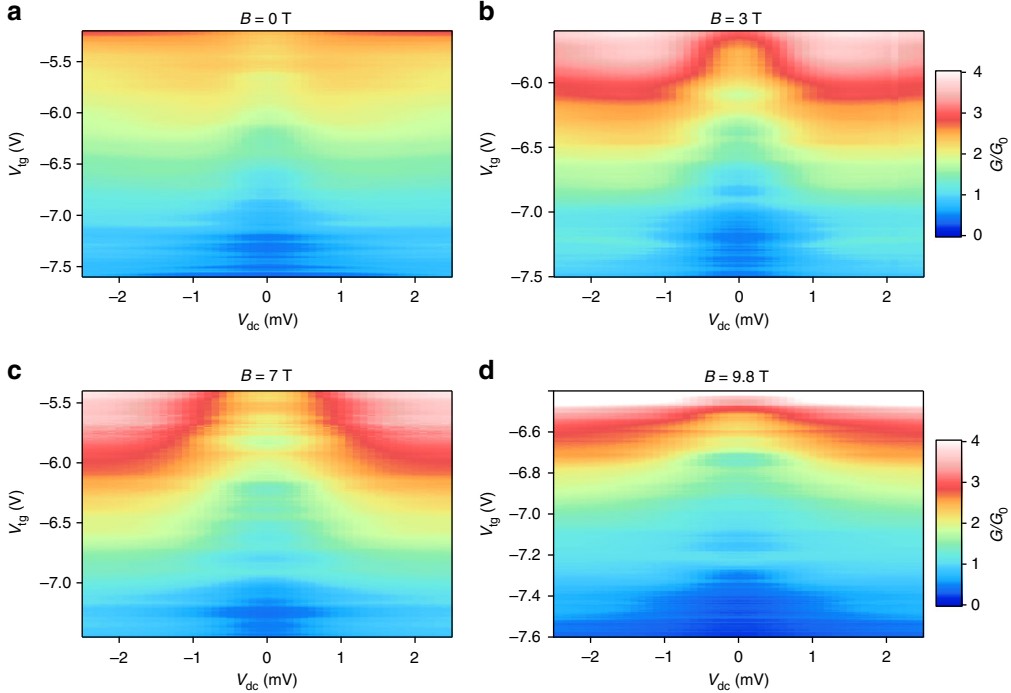

**Fig. 4** Color maps of the differential conductance $G$ as a function of the applied offset $V_{dc}$ and the split-gate voltage at different magnetic fields. **a** $V_{tg}$ was lowered by 20 mV between successive $V_{dc}$ sweeps from −5.2 to −7.6 V, $B = 0$ T. **b** $V_{tg}$ was decreased by 15 mV between successive $V_{dc}$ sweeps from −5.6 to −7.5 V. A magnetic field of $B = 3$ T was applied perpendicularly to the 2DEG. **c** $V_{tg}$ was lowered by 15 mV for successive $V_{dc}$ sweeps from −5.5 to −7.45 V. A magnetic field of $B = 7$ T was applied perpendicularly to the 2DEG. **d** $V_{tg}$ was decreased by 9 mV between successive $V_{dc}$ sweeps from −6.55 to −7.6 V. A magnetic field of $B = 9.8$ T was applied perpendicularly to the 2DEG

from the combination of wiring, contact resistance between metal electrodes and graphite as well as between graphite contacts and single-layer MoS$_2$, and finally the geometrical resistance of the MoS$_2$ sheet on both sides of the constriction. As a result, we observe eight consecutive, regular steps appearing at wholes and halves of $G_0 = 2e^2/h$, which indicates the lifting of all degeneracies in our sample. Slight deviations from perfect multiples of $e^2/h$ might be due to variation in the resistance of the reservoirs around the constriction originating from the presence of charged impurities[21,22]. In monolayer MoS$_2$, there is spin splitting near the conduction band minimum of MoS$_2$, while the K and K′ valleys are degenerate[3,4,10]. Our measurement implies that inside the constriction area, the valley degeneracy is lifted and therefore the conductance quantizes into singular spin states. We note that such conductance quantization is also observed in other prepared devices (See Supplementary Fig. 2 and Supplementary Note 2). Similarly to this device, we observe the conductance steps at multiplies of $0.5 \times G_0$ corroborating the lifting of all degeneracies in monolayer QPCs. Such valley degeneracy lifting in a QPC geometry is not unique to monolayer MoS$_2$. Similar response was observed in Si–SiGe heterostructures[23,24], graphene[25], and carbon nanotubes[26].

**Bias spectroscopy**. We next perform bias spectroscopy measurements to investigate the 1D sub-band energy spacing inside our monolayer MoS$_2$ QPC. For this purpose, we measure the differential conductance $G = dI/dV$ using the lock-in technique with a small ac signal at finite dc source-drain bias $V_{dc}$ and at different fixed values of $V_{tg}$. The resulting conductance variation is plotted as color maps at different magnetic fields in Fig. 4a–d. We can clearly see the regions of quantized conductance around $V_{dc} = 0$ mV and how it saturates at higher offset values. In order

to understand in more detail the evolution of quantized values, we also represent the data as line maps in Fig. 5. We first discuss the measurements performed at $B = 0$ T (Fig. 5a). In this map, the conductance plateaus appear as dark regions with increased density of line traces. Note that the lower the trace is in the map, the more negative is the applied top-gate voltage $V_{tg}$. In the center ($V_{dc} = 0$ mV), we observe the bunching of lines in the range $0.5–2.5 \times G_0$ at regular spacing of $0.5 \times G_0$ after subtraction of the background resistance contribution (see Supplementary Information). All presented maps were symmetrized following the model of Kristensen et al.[27] (see Supplementary Fig. 3 and Supplementary Note 3). We follow the increase of the differential conductance by continuously increasing the dc bias voltage. At $V_{dc} \sim 0.8$ mV, we observe a saturation of the differential conductance at the corresponding value of a half plateau at about $\left(n - \frac{1}{2}\right) \times \frac{1}{2} G_0$, as expected for the adiabatic transport model in QPCs[28]. We attribute any deviation from ideal half-plateau conductance values to the fact that we study a very low number of subbands $n$ just above the band gap of MoS$_2$. On the contrary, for higher subbands like e.g. $2G_0$, the traces evolve only up to the expected half plateau values $(2.25G_0)$ in good agreement with the proposed model.

In Fig. 5b–d we present the same bias spectroscopy maps taken at constant perpendicular magnetic fields of 3, 7, and 9.8 T. While qualitatively very similar, these maps allow us to quantitatively measure the evolution of the spin splitting and Zeeman energy between the subbands of the QPC. Similarly to observations in other 2DEG systems like GaAs/AlGaAs[29] at higher magnetic fields, the discrete conduction levels at $V_{dc} = 0$ V and their evolution to half plateaus become better visible, as a smaller number of intermediate lines are present in the maps. The transition to the half-plateau values in these maps occurs at

consecutively higher dc bias values as the magnetic field increases, indicating the continuous increase of Zeeman energy splitting. From the evolution of the Zeeman energy as a function of the magnetic field, we can extract the electron $g$-factor in the conduction band of $MoS_2$.

In Fig. 6a, we present normalized representative curves for the $V_{dc}$ dependence of the differential conductance at different constant magnetic fields. In this direct comparison, it is apparent how the splitting energy between the opposite spin levels is increasing with increasing magnetic field. We extract the sub-band

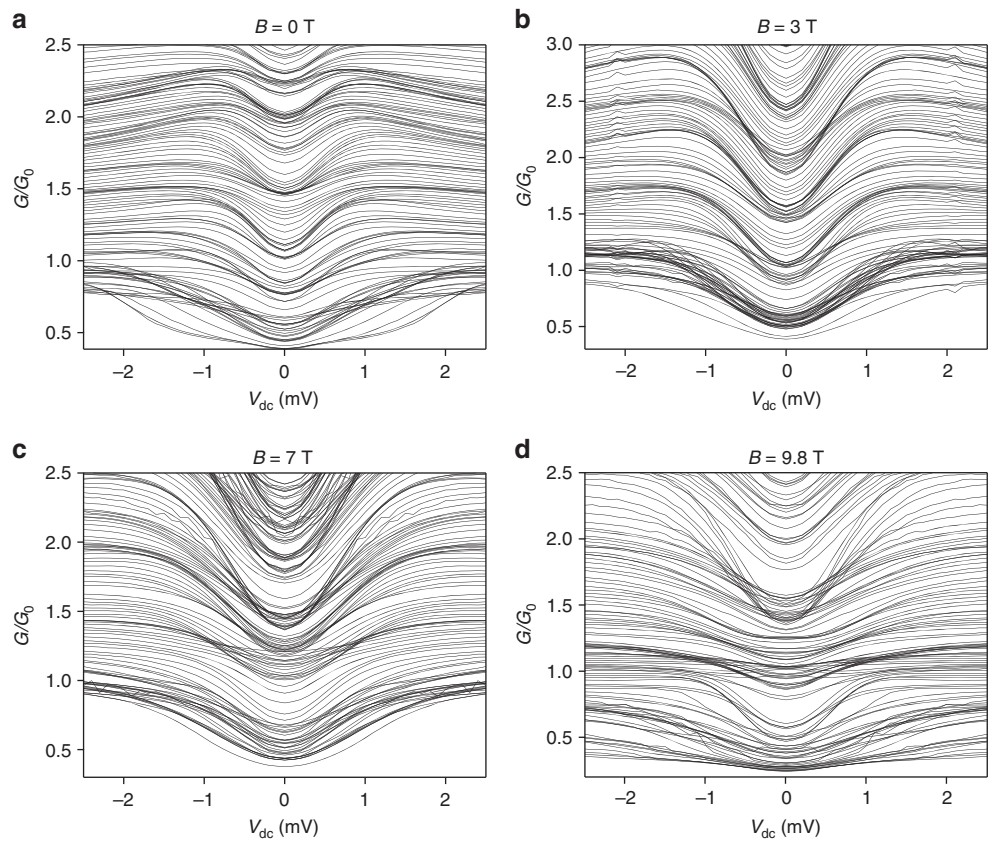

**Fig. 5** Traces of the differential conductance $G$ as a function of the applied $V_{dc}$ for different split-gate voltages. **a** The split-gate voltage was decreased by 20 mV between successive $V_{dc}$ sweeps from −5.16 to −7.6 V. No magnetic field was applied. **b** The split-gate voltage was decreased by 15 mV between successive $V_{dc}$ sweeps from −5.64 to −7.5 V. A magnetic field of 3 T was applied perpendicularly to the 2DEG. **c** The split-gate voltage was decreased by 15 mV between successive $V_{dc}$ sweeps from −5.5 to −7.45 V. A magnetic field of 7 T was applied perpendicularly to the 2DEG. **d** The split-gate voltage was decreased by 9 mV between successive $V_{dc}$ sweeps from −6.548 to −7.7 V. A magnetic field of 9.8 T was applied perpendicularly to the 2DEG

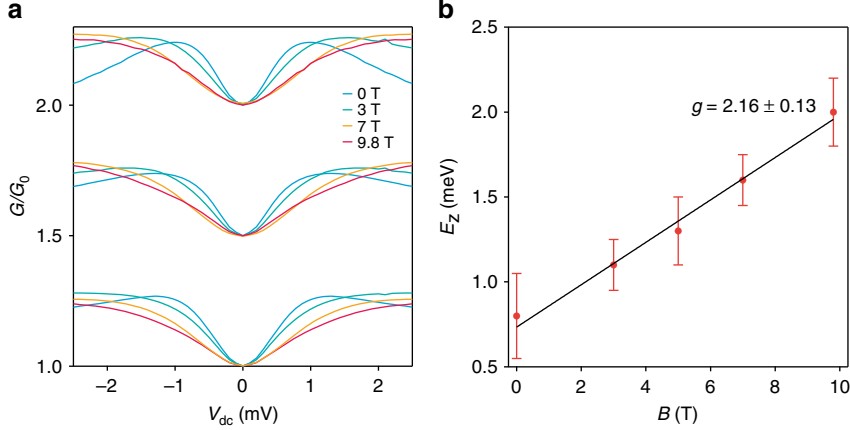

**Fig. 6** Magnetic field dependence and the $g$-factor. **a** Normalized curves from the maps shown on Fig. 4 in the range of 1 to 2.5 × $G_0$ showing the continuous increase in the Zeeman energy $E_z$ with the strength of the applied magnetic field. **b** Linear fit of the Zeeman energy $E_z$ as a function of the magnetic field and the extraction of the Landé $g$-factor in the conduction band of monolayer $MoS_2$. The error bars correspond to the s.e.m. of the data and were extracted from the statistical deviation of the estimated saturation values at the different conductance transition in the range 0.5–2 × $G_0$ in each map from Fig. 5

Zeeman splitting energy values from the points at which the curves reach saturation or at higher magnetic fields, the $V_{dc}$ value where the slope changes. In Fig. 6b we plot the thus extracted Zeeman energy as a function of the magnetic field taking into account the uncertainty of the half plateau saturation for different conductance steps from all the bias spectroscopy maps. The Zeeman energy follows a linear dependence. The slope of this line can be fitted using $\Delta E_z/B = |g|\mu_B$[30,31], where $\mu_B$ is the Bohr magneton. In this way, we extract a g-factor of $2.16 \pm 0.13$ in the conduction band of MoS$_2$. This value is in good agreement with DFT calculations predicting a g-factor of about 2.2[4]. This indicates very low electron-electron interaction in single-layer MoS$_2$-QPC. We also note that the extracted value is very similar to the g-factor of a free electron, as well as the g-factor in other two-dimensional materials like graphene[32] and black phosphorus[33].

## Discussion

We present the measurement of the spin-splitting in the conduction band of single-layer MoS$_2$ and its evolution in magnetic field. Our high electron mobility device allows us to access to the intrinsic properties of electrons in the conduction band of MoS$_2$. By applying a large positive voltage to the back gate we achieve high homogeneous doping of the channel. Using the top gate electrodes, we locally deplete the MoS$_2$ sheet and form a constriction for the electrons, a QPC. We observe quantization of the conductance in multiples of $e^2/h$ revealing lifted spin and valley degeneracy. Performing bias spectroscopy at different magnetic fields, we extract the spin splitting and g-factor values in the conduction band of monolayer MoS$_2$. The direct resolution of spin splitting on the order of meV which can be further enhanced by bringing single-layer MoS$_2$ in close proximity to the magnetic insulator substrate[34] could pave the way for novel 2D spintronic devices.

## Methods

**Material transfer and device fabrication**. Bottom h-BN flakes were directly exfoliated on Si substrates covered with 270 nm thermally grown SiO$_2$. Thin flakes were identified using optical microscope and AFM. All further flakes were exfoliated on PDMS substrates. Single-layer MoS$_2$ was identified by optical contrast and further confirmed by dark field microscope PL measurements. Flakes were aligned and transferred on the target substrate in a home-built transfer station with micromanipulators. The exfoliated flake on the inverted PDMS stamp was aligned and brought into contact with the target substrate, which is heated to up to 70 °C for better adhesion. The stamp is cooled down to room temperature and slowly lifted from the substrate, resulting in the transfer of the flake onto the target substrate (see Supplementary Fig. 4 and Supplementary Note 4).

After the complete stack is deposited onto SiO$_2$, the wafer is annealed for 8 h at 360 °C in high vacuum in order to improve the adhesion between the layers and remove residues from the transfer. Source-drain electrodes and top gates are defined by conventional e-beam lithography followed by the e-beam deposition of Ti/Au (2/85 nm) electrodes. A final annealing at 100 °C for 8 h inside the measurement chamber at a pressure of $5 \times 10^{-6}$ mbar prior to characterization is performed in order to improve the contact resistance between metals and FLG and to remove fabrication residues.

**Electrical transport measurements**. Electrical characterization is carried out using a National Instruments virtual DAQ lock-in amplifier, a Basel physics LSK389A current amplifier and a Keithley 2636 sourcemeter as a DC voltage source. Cryogenic measurements were performed in an ICE Oxford liquid helium continuous flow cryo-magnetic system with a base temperature of 1.4 K. To avoid heating up of the sample and charging, we used an ac excitation with an amplitude of 100 μV and a frequency of 13 Hz. Gate leakage currents were kept as small as possible, generally lower than 50 pA.

**Data availability**. The data that support the findings of this study are available from the corresponding author on reasonable request.

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

## Acknowledgements

We thank Klaus Ensslin, Adrien Allain, Dmitry Ovchinnikov and Dmitrii Unuchek for fruitful discussions and help. Devices were fabricated in the EPFL Center for Micro/Nanotechnology (CMI). We acknowledge the help of Z. Benes (CMI) with e-beam lithography. K.M., A.A., and A.K. would like to acknowledge support by the European Research Council (ERC, Grant 682332), Marie Curie ITN network "MoWSeS" (grant no. 317451), Swiss National Science Foundation (Grant 157739), and Marie Curie-Sklodowska COFUND (grant 665667). A.K. acknowledges funding from the European Union's Horizon 2020 research and innovation program under grant agreement No 696656 (Graphene Flagship). K.W. and T.T. acknowledge support from the Elemental Strategy Initiative conducted by the MEXT, Japan and JSPS KAKENHI Grant Numbers JP15K21722 and JP25106006.

## Author contributions

K.M., A.A., and A.K. designed the experiments. K.M. and A.A. fabricated the samples. K.M. performed the measurements. K.W. and T.T. grew the h-BN crystals. K.M, A.A., and A.K. wrote the manuscript. All authors discussed the results.

## Additional information

**Competing interests:** The authors declare no competing financial interests.

