## [Peer Review File · Nature Communications]

Reviewers' comments:

Reviewer #1 (Remarks to the Author):

Authors have investigated spin splitting in conduction band in monolayer MoS₂, through the fabrication of Quantum Point Contacts and the experimental observation at cryogenic temperatures of the occupation of the different conduction band states.

I believe that the present paper is timely and really interesting and certainly deserves the publication on Nature Communications.

I have only really minor remarks.

Figure 3 shows the normalized conductance as a function of the split gates biases for one single device. I believe that, since such measurements are a little bit noisy, in order to corroborate the fact that indeed quantum conductance plateaus are observed, I would suggest to show G-V curves for other devices.

I believe that Fig. 4 would be improved in terms of readability, if it was provided through a colormap.

Reviewer #2 (Remarks to the Author):

The manuscript reports the analysis of conduction band (spin-)splitting in monolayer MoS₂ by electrical transport measurements.

Prior work has demonstrated spin splitting in the valence band of this material, which is easier to observe because the splitting is larger and the occupied states can be probed by photoemission. The conduction bands have been studied for doped samples in ARPES, while splitting was not demonstrated.

In the present work, the conductance steps at half integer G_0 values indicate the lifting of all degeneracies (including spin degeneracy) in the setup. This has been further investigated by bias-voltage spectroscopy to extract the Zeeman energies.

I believe the manuscript to be suitable for publication in Nature Communications. I have only minor comments.

The error bar in Figure 5b seems to underestimate the variation in the plateau energies, in particular for zero field, the plateaus seem to be above 1mV.

The summary/conclusions part could be shortened.

Referee 1:

Comment: Authors have investigated spin splitting in conduction band in monolayer MoS₂, through the fabrication of Quantum Point Contacts and the experimental observation at cryogenic temperatures of the occupation of the different conduction band states.

I believe that the present paper is timely and really interesting and certainly deserves the publication on Nature Communications.

Reply: We thank the reviewer for his/her time spent for reviewing our manuscript. We are glad that he/she finds the results timely and interesting.

Comment: I have only really minor remarks. Figure 3 shows the normalized conductance as a function of the split gates biases for one single device. I believe that, since such measurements are a little bit noisy, in order to corroborate the fact that indeed quantum conductance plateaus are observed, I would suggest to show G-V curves for other devices.

Reply: We understand the concern of the reviewer. We present a qualitatively similar device showing quantization results, which endorse our finding that all degeneracies are lifted in a QPC in monolayer MoS₂. It was fabricated in the same manner as the one in the main text. The results shown in Fig. R1 is added in the revised Supplementary information and mentioned in the main text.

Figure R1. Observation of conductance quantization in another similar device. **a**, AFM image of the split gate geometry, which is very similar to that of the device in the main text. **b**, Height profile along the blue line displayed in **a**. The distance between the two top gates is about 100 nm. **c**, Conductance of the device as function of the applied top-gate voltage after subtraction of series resistance of 8.3 k Ω . Clear quantization steps in whole and half values of G_0 are observed.

Comment: I believe that Fig. 4 would be improved in terms of readability, if it was provided through a colormap.

Reply: We thank the reviewer for his/her recommendation. In order to improve the readability of the figure we added a new figure showing the data as color map in Fig.4. This new figure is shown below for the reviewer's attention.

Figure R2. Color maps of the differential conductance G versus the applied offset V_{dc} and the split gate voltage at different magnetic fields. (a) V_{tg} was lowered by 20 mV between successive V_{dc} sweeps from -5.2 V to -7.6 V, $B = 0$ T. (b) V_{tg} was decreased by 15 mV between successive V_{dc} sweeps from -5.6 V to -7.5 V. A magnetic field of $B = 3$ T was applied perpendicularly to the 2DEG. (c) V_{tg} was lowered by 15 mV for successive V_{dc} sweeps from -5.5 V to -7.45 V. A magnetic field of $B = 7$ T was applied perpendicularly to the 2DEG. (d) V_{tg} was decreased by 9 mV between successive V_{dc} sweeps from -6.55 V to -7.6 V. A magnetic field of $B = 9.8$ T was applied perpendicularly to the 2DEG.

Referee 2:

Comment: The manuscript reports the analysis of conduction band (spin-)splitting in monolayer MoS₂ by electrical transport measurements. Prior work has demonstrated spin splitting in the valence band of this material, which is easier to observe because the splitting is larger and the occupied states can be probed by photoemission. The conduction bands have been studied for doped samples in ARPES, while splitting was not demonstrated. In the present work, the conductance steps at half integer G_0 values indicate the lifting of all degeneracies (including spin degeneracy) in the setup. This has been further investigated by bias-voltage spectroscopy to extract the Zeeman energies.

I believe the manuscript to be suitable for publication in Nature Communications.

Reply: We are very glad that the reviewer recognizes the significance of our work and recommends publication.

Comment: The error bar in Figure 5b seems to underestimate the variation in the plateau energies, in particular for zero field, the plateaus seem to be above 1mV.

Reply: After the reviewer comment, we payed closer attention to the uncertainty in the maps and acknowledge the comment of the referee. It is now corrected. The updated figure is shown below and included in the revised manuscript.

Figure R2. Magnetic field dependence and the g-factor. (a) Normalized curves from the maps shown on figure 4 in the range of 1 to $2.5 \times G_0$ showing the continuous increase in the Zeeman energy E_z with the strength of the applied magnetic field. (b) Linear fit of the Zeeman energy E_z as a function of the magnetic field and the extraction of the Landé g-factor in the conduction band of monolayer MoS₂.

Comment: The summary/conclusions part could be shortened.

Reply: We welcome this suggestion. This section is shortened in the revised manuscript as follows:

In conclusion, we present the measurement of the spin-splitting in the conduction band of single-layer MoS₂ and its evolution in magnetic field. Our high electron mobility device allows us to access to the intrinsic properties of electrons in the conduction band of MoS₂. By applying a large positive voltage at the back gate we achieve high homogeneous doping of the channel. Using the top gate electrodes, we locally deplete the MoS₂ sheet and form a constriction for the electrons, a quantum point contact. We observe quantization of the conductance in multiples of e^2/h revealing lifted spin and valley degeneracy. Performing bias spectroscopy at different magnetic fields, we extract the spin splitting and g-factor values in the conduction band of monolayer MoS₂. The direct resolution of spin splitting value in the order of meV which can be further enhanced by bringing the single-layer MoS₂ in a close proximity to the magnetic insulator substrate³⁴ could pave the way for novel 2D spintronics devices.

REVIEWERS' COMMENTS:

Reviewer #1 (Remarks to the Author):

Authors have addressed my previous issues. I believe that the paper in the present form deserves publication in Nat. Comm.

Reviewer #2 (Remarks to the Author):

The authors have responded to my suggestions and have included minor changes. The manuscript can be published.

Our replies to referees

Dear reviewers,

We are very delighted about your positive comments and acceptance of our manuscript.

Referee 1:

Comment: Authors have addressed my previous issues. I believe that the paper in the present form deserves publication in Nat. Comm.

Reply: We thank the reviewer for his/her kind support in improving our manuscript. We are glad that she/he recommends publication.

Referee 2:

Comment: The authors have responded to my suggestions and have included minor changes. The manuscript can be published.

Reply: We appreciate the reviewer's suggestions and are happy that she/he is satisfied with the revised manuscript.